# Assessment of Phenotypic Characteristics, Polysaccharide Composition, and Hypoglycemic Potential in Different Commercial Grades of *Lycium barbarum*: A Comprehensive Study Using HPLC and NMR

**DOI:** 10.3390/foods14223862

**Published:** 2025-11-12

**Authors:** Caixia Ma, Fei Liu, Linwu Ran, Jia Mi, Lu Lu, Siyu Wang, Xinyu Ge, Bo Jin, Lutao Zhang, Yamei Yan

**Affiliations:** 1School of Food and Biological Engineering, Hefei University of Technology, Hefei 230601, China; mcx2015@163.com; 2Institute of Wolfberry Science, Ningxia Academy of Agriculture and Forestry Sciences, Yinchuan 750002, China; 3Nutrition and Bromatology Group, Department of Analytical Chemistry and Food Science, Faculty of Science, University of Vigo, E32004 Ourense, Spain; 4School of Public Health, Ningxia Medical University, Yinchuan 750004, China; 5School of Food Science and Engineering, Ningxia University, Yinchuan 750021, China; 6College of Pharmacy, Ningxia Medical University, Yinchuan 750004, China

**Keywords:** *Lycium barbarum* L. (*L. barbarum*), phenotypic characteristics, polysaccharides, nuclear magnetic resonance, quality evaluation

## Abstract

*Lycium barbarum* L. (abbreviated to *L. barbarum*), a traditional dual-use plant as food and medicine, contains polysaccharides from *Lycium barbarum* L. (LBPs) as its key bioactive component. This study aimed to examine the phenotypic characteristics, polysaccharide content, and their correlation with activity across various commercial grades of *L. barbarum*. Five commercial grades of *L. barbarum* were selected for analysis to determine their phenotypic characteristics and polysaccharide content. High-performance liquid chromatogram-diode array detection (HPLC-DAD) and ^1^H NMR were employed to analyze the monosaccharide composition of LBPs, of which their hypoglycemic activity was further valuated. Results revealed significant differences in fruit weight and diameter among different grades (*p* < 0.05), while floating rate and bulk density remained unaffected by grades. Variations were observed in the chromaticity coordinates, with the *c* values showing notable differences (*p* < 0.01). Polysaccharide content tended to increase with higher grades and smaller fruit sizes, ranging from 1.94% to 5.69%. The polysaccharides in different contained monosaccharides of Man, Rha, Ara, Gal, Glc, GalA, GlcA and Xyl, with Ara and Gal being predominant. Identified through ^1^H NMR spectra, the peak intensity of Ara increased from lower to higher grades, and the arrangement of the chemical shifts reflected distinct commercial grade characteristics. The inhibitory concentration (IC_50_) against *α*-amylase and *α*-glucosidase ranged from 0.418 to 1.345 mg/mL, and 0.474 to 1.052 mg/mL, respectively, indicating good hypoglycemic activity within this range. The main monosaccharide groups Ara, Gal, and GalA were identified as key contributors to enzyme inhibition. Collectively interpreting the phenotypic features, polysaccharide content, monosaccharide composition, NMR data and activity profiles, Ara, Gal and GalA emerge as signature monosaccharide components of LBPs. These results provide novel theoretical insights for *L. barbarum* quality assessment.

## 1. Introduction

*L. barbarum* (Wolfberry; Goji), an ancient deciduous shrub belonging to the Solanaceae family, is widely recognized as a source of medicine and food [1]. Valued for its exceptional drought tolerance and resistance to salt and alkali, it has a broad distribution across regions in China, including Ningxia, Qinghai, Gansu, and Xinjiang [2]. As a traditional Chinese medicinal material, *L. barbarum* demonstrates diverse biological activities, including antioxidant, antitumor [3], blood glucose-reducing [4], and vision/liver protective properties [5]. Traditional quality assessment primarily centers on fruit attributes such as size, shape, color, texture, and taste, with larger, brightly red, intact, and sweet fruits commanding higher commercial appeal. China National Standard GB/T 18672-2014 “Wolfberry” [6] categorize fruits into four grades based on quantity per50 g: Extra Superior (≤280 fruits), Superior (≤370), Grade A (≤580), and Grade B (≤900), with larger fruits typically receiving superior grades (Appendix A). Despite this grading system, contemporary market dynamics reveal the inadequacy of conventional evaluation criteria for comprehensive quality assessment. To enhance insights into quality characteristics and determining factors, researchers have explored the impact of genotype and agricultural practices on fruit attributes. Kafkas’ comparative studies on three *Lycium* genotypes unveiled significant genotypic effects on fruit dimensions and chromatic values [7]. Wan’s examination on seven agronomic traits under potassium fertilization demonstrated potassium’s influence on the longitudinal to transverse diameter ratio in fresh fruits, with limited impact on other traits [8]. Current research efforts are increasingly focusing on quality evaluation through the analysis of functional components.

*Lycium barbarum* polysaccharides (LBPs) are a primary active component of the fruit and are a major quality marker in both the Chinese Pharmacopoeia (CP) and International Organization for Standardization (ISO) standards [9]. LBPs are considered one of the main components that enable the biological activity along with carotenoids, polyphenols, vitamins, and minerals. These water-soluble complexes mainly consist of polysaccharide chains, which are composed of eight monosaccharides: arabinose (Ara), galactose (Gal), galacturonic acid (GalA), glucose (Glc), glucuronic acid (GlcA), rhamnose (Rha), mannose (Man), and xylose (Xyl), along with conjugated glycoproteins [10]. Recent studies has frequently delved into quality assessment through the physicochemical analysis of polysaccharides and their bioactivity properties. Cui established a set of quality evaluation criteria for *L. barbarum* sourced from various geographical regions by quantifying the polysaccharide content across forty samples [11]. In a similar vein, Liu developed a regional forecasting model for *L. barbarum* samples by generating high-performance liquid chromatography (HPLC) fingerprint profiles of their polysaccharides [12]. However, the correlation between commercial *L. barbarum* grades and the content, composition, and bioactivity of their functional components has not been sufficiently investigated.

In this study, commercially accessible *L. barbarum* fruits were classified into five grades in accordance with the granularity values stipulated by the China National Standard GB/T 18672-2014, titled “Wolfberry” [6]. The study assessed phenotypic characteristics and polysaccharide content across various grades. Subsequently, the monosaccharide composition of LBPs was analyzed employing high-performance liquid chromatography with diode array detection (HPLC-DAD) and proton nuclear magnetic resonance (^1^H NMR) spectroscopy. The in vitro hypoglycemic activity of crude LBPs was also investigated. By examining the correlations between commercial grades and phenotypic traits, functional component profiles, and bioactivity, this research aims to establish a theoretical foundation for enhancing quality assessment systems and optimizing the utilization of *L. barbarum*, there by maximizing their dual medicinal and nutritional value.

## 2. Materials and Methods

### 2.1. Materials and Reagents

A total of 48 dried *L. barbarum* fruit samples were collected from the Goji Trading Market, Zhongning, China. Enzymes (minimum purity 99.5%), was purchased from Solarbio (Beijing, China). Standards (mannose, glucuronic acid, galacturonic acid, xylose, galactose, arabinose, and glucose) were purchased from Sigma Chemical Co., Ltd. (St. Louis, MO, USA). HPLC-grade acetonitrile was purchased from TEDIA Co., Inc. (Fairfield, IA, USA). 1-phenyl-3-methyl-5-pyrazolone (PMP) was purchased from Gibco (Carlsbad, CA, USA). All organic solvents used for separation were of domestic analytical grade, trifluoroacetic acid (TFA), 4-Nitrophenyl-*α*-D-glucopyranoside (*p*NPG), phenol, and concentrated sulfuric acid were provided by Damao.

### 2.2. Classification of L. barbarum Samples Based on Refined Commercial Grades

The *L. barbarum* specimens selected for this investigation have been stratified into four categories in accordance with the GB/T 18672-2014 Standard [6], as previously referenced, based on the quantity of grains per 50 g. Although these classifications provide an initial evaluation of quality, significant variations in quality persist within specific grain size categories. Consequently, to more precisely delineate the quality distinctions among fruits of varying grain sizes, there exists a necessity to refine the grading system by further subdividing the grain size ranges. Hence, this study introduces an enhanced system, as outlined in Table 1, which classifies the berries from One to Five in accordance with grain sizes. The proposed methodology preserves the foundational structure of the original grading system and augments it to more accurately reflect the quality differences and market demands of fruit.

### 2.3. Measurement of Phenotypic Characteristics

The longitudinal diameter, and transverse diameter of *L. barbarum* were assessed using a MICROTEK scanner (Yiheng Technology Co., Ltd., Shanghai, China) and Wan shen Seed Analysis Software (V2.1.4.7, Wanshen Testing Technology Co. Ltd., Hangzhou, China). The color parameters, including *L** (brightness), *a** (red-green value), *b** (yellow-blue value), *c* (saturation), and *h* (chromaticity angle), were measured using a CM-5 spectrophotometer (Konika Minolta, Japan) [13]. The floating rate was determined according to the method described by Gong [14], while the bulk density was measured following the method outlined by He [15]. To ensure consistency and reliability of the data, each sample was measured in triplicate.

### 2.4. Extraction and Quantification of Polysaccharides

Extraction of Crude Polysaccharides. *L. barbarum* contains a certain proportion of crude fat. Given the outstanding extraction efficiency, low toxicity, low boiling point, and ease of recovery, petroleum ether is utilized as the preferred extraction reagent for crude fat removal [16]. Crude polysaccharides were extracted using the method developed by Redgwell [17].

Quantification of Polysaccharides Content. The polysaccharide content was determined using the phenol-sulfuric acid method, as specified in the 2020 edition of the Chinese Pharmacopoeia Part I [18]. In summary, an appropriate amount of the *L. barbarum* extract was diluted with distilled water to a volume of 2.0 mL. Then, 1.0 mL of 5% phenol solution was added, followed rapidly by 5.0 mL of concentrated sulfuric acid. The sample was thoroughly mixed, allowed to stand for 5 min, and then placed in a boiling water bath for 15 min. Then, the sample was cooled to room temperature. A blank control was prepared by adding 2.0 mL of distilled water to the same procedure. The absorbance was measured at 490 nm. The polysaccharide content was calculated using the standard substance method.

Following sample determination, the remaining extract was concentrated through rotary evaporation, dialyzed with a 5000 Da molecular weight cutoff membrane, and freeze-dried to obtain crude LBPs.

### 2.5. Monosaccharide Composition Analysis

The monosaccharide composition was determined by HPLC by the described method of pre-column derivatization with PMP [19]. The specific methods were as follows: Firstly, the monosaccharide standards were mixed into a mixed aqueous solution (the concentration of each monosaccharide was 10 mg/mL). 50 μL of the mixed standard monosaccharide solution was pipetted into 50 μL 0.6 M NaOH solution and mixed well. Then the mixture (100 μL) was labeled with PMP by adding 100 μL of 0.5 M methanol solution of PMP. The following reaction took place in a 70 °C constant-temperature blast oven for 100 min, at which point it was then removed and cooled down. 0.3 M hydrochloric acid (100 μL) was added for neutralization and then evaporated. Next, the extraction was performed three times with chloroform. The aqueous layer was filtered through a 0.22 μm membrane and analyzed by HPLC.

To the polysaccharide sample solution (100 μL, 10 mg/mL) was added 4 M TFA, 100 μL to hydrolyze at 110 °C for 6 h. After hydrolysis, methanol solution (200 μL) was added to remove excess TFA, and then evaporated at reduced pressure. This procedure was repeated again to remove the TFA. The residue was dissolved in 100 μL of 0.3 M NaOH solution. Then 100 μL of 0.5 M methanol solution of PMP was added to the mixed solution and the reaction was carried out at 70 °C for 100 min. After cooling, the solution was neutralized, extracted, and filtered using a 0.22 μm membrane in the same manner as that used for the mixed standard monosaccharide solution.

Chromatographic conditions were set as follows: Mobile phase: *V*_PBS_:*V*_acetonitrile_ = 83:17; Chromatographic column: Agilent 959990-902 C_18_, dimensions 4.6 × 250 mm; flow rate: 1.0 mL/min; UV detection wavelength: 250 nm; injection volume: 20 μL; column temperature: 35 °C.

### 2.6. ^1^H NMR Analysis

10 mg of dried LBPs powder was dissolved in deuterium oxide (D_2_O) at a concentration of 10 mg/mL. In order to enhance the visibility of the carbohydrate signals, the samples were treated thrice with heavy water dissolution to perform the hydrogen-deuterium exchange and a following freeze-drying procedure for solvent removal. Subsequently, 5 mg of the resultant solid was dissolved in 0.6 mL of D_2_O and transferred to an NMR tube. The sample was permitted to settle for seven days to facilitate the precipitation of undissolved LBPs solids, yielding a clear upper-layer solution. One-dimensional ^1^H NMR spectra of LBPs from various commercial grades were obtained using an Avance NEO 400 MHz spectrometer (Bruker, Germany) (*T* = 300 K; 500 scans) [20]. For quantitative comparisons, integrated peak areas were used.

Due to the limited solubility of the sample, the protons exhibit reduced signals, which affect solvent correction and proton assignment. Consequently, the ethanol impurity peak (*δ*1.17, ^3^*J*_HH_ = 7.1 Hz, triplet, C*H*_3_CH_2_OH) present in all spectra was utilized to correct the chemical shift in the spectra. The polysaccharides peaks were screened using the MNova software (V14.0.0) to generate spectra with solvent and impurities shielded. It is important to note that this analysis specifically targets polysaccharide-derived components; free sugars like fructose were removed during the dialysis and purification process and are therefore not detected. Future studies could employ sample pre-treatment or alternative solvents to improve solubility for 2D NMR analysis.

### 2.7. In Vitro Hypoglycemic Activity Assessment

*α*-Amylase Inhibition Assay. The methodology delineated by He was referenced and subsequently adapted [21]. Crude polysaccharide solutions were prepared at concentrations of 0.1, 1, 2, 3, and 5 mg/mL. Each solution was subsequently dispensed in 100 μL aliquots into centrifuge tubes. To these, 100 μL of a 0.1 mol/L phosphate-buffered saline (PBS) solution at pH 6.8 and 100 μL of a 2 U/mL *α*-amylase solution were introduced. The resultant mixtures were subjected to incubation at 37 °C for a duration of 10 min. Following this, 100 μL of a 1% soluble starch solution was introduced, and the reactions were allowed to continue at 37 °C for an additional 10 min. Subsequently, 300 μL of DNS reagents were added to each tube. The reactions were terminated by boiling the samples in water for 10 min, after which the absorbance was measured at a wavelength of 540 nm. A phosphate-buffered saline solution served as the negative control, whereas acarbose functioned as the positive control. The inhibition rate and the IC_50_ value were subsequently calculated.

*α*-Glucosidase Inhibition Assay. The method was modified based on the methodology by Kim [22]. The crude LBPs was formulated into solutions at concentrations of 0.1, 1, 2, 3, and 5 mg/mL, respectively. Subsequently, 50 μL of each solution was dispensed into a centrifuge tube. Thereafter, 50 μL of 0.1 mol/L PBS solution (pH 6.8), 100 μL of 2 U/mL *α*-glucosidase solution, and 50 μL of 5 mmol/L pNPG were introduced. The resultant mixture was subjected to incubation at 37 °C for a duration of 20 min. Upon completion of the reaction, 100 μL of 1.0 mol/L Na_2_CO_3_ solution was introduced to terminate the reaction. The absorbance data was procured at a wavelength of 405 nm. PBS served as the negative control, whereas acarbose functioned as the positive control. The inhibition rate and the IC_50_ value of *α*-glucosidase were ascertained.

### 2.8. Data Processing

The data were presented as the mean derived from three replicate measurements. Analysis of variance was employed to ascertain the differences among the outcomes. The Duncan test was utilized for the comparison of datasets. A value of *p* < 0.05 was considered to indicate a statistically significant difference, while a value of *p* < 0.01 was deemed to represent a highly significant difference. The IC_50_ values were computed utilizing SPSS version 27.0. The graphs were subjected to analysis through the use of Origin 2021 software. Partial least squares analysis was executed using the SIMCA14.0 software package.

## 3. Results and Discussion

### 3.1. Phenotypic Analysis of Various L. barbarum Fruit Grades

Key parameters such as single fruit weight, diameter, floating rate, bulk density, and color are crucial in assessing the quality of *L. barbarum* fruits. The floating rate indicates water content and maturity; higher rates suggest more advanced maturity and lower water content. Bulk density is a key indicator that reflects both yield and quality; high density correlates with compactness and fullness, whereas low density suggests looseness or the presence of impurities [23]. Color is a significant factor that influences consumer choice and directly impacts the product’s commercial value.

Figure 1A–F illustrates that the average weight of a single fruit across various grades is 0.15 ± 0.07 g, with the average longitudinal and transverse diameters being 12.80 ± 1.93 mm and 6.09 ± 1.26 mm, respectively. Significant differences (*p* < 0.05) were observed in the weight and dimensions of single fruits among grades, which is consistent with previous research findings [24]. The floating rate ranged from 55.56% to 100%, and the bulk density varied from 0.42 to 0.56 g/mL, indicating limited differences. Color parameters were consistent across grades, with *L** values between 29.67 and 35.00, *a** values from 31.33 to 42.67, *b** values from 29.00 to 41.60, and *h* values from 43.87 to 49.99, showing no significant differences among grades. However, the *c* value, which reflects color saturation, exhibited a notable range of 44.09 to 58.35 (Appendix A), with a significant difference (*p <* 0.01) among grades. Lower-grade fruit (with bigger sizes and lower grain counts per 50 g) had higher *c* values, indicating greater color saturation, corroborating previous research by Kafkas [7].

### 3.2. LBPs Content Analysis

As indicated in Table 2, the LBPs content across various grades varies from 1.94% to 5.69%, exceeding the national standard minimum of 1.8%. Typically, fruits with higher grade numbers and smaller sizes exhibit higher polysaccharide levels, which is consistent with previous findings by Gao [25]. However, the average LBPs content increases in the sequence: 5 < 1 < 2 < 3 < 4. The relatively lower polysaccharide content in Grade 5 is a notable exception, potentially due to differences in accumulation patterns during fruit maturation [26]. Overall, assessing *L. barbarum* fruit quality based on polysaccharide content appears to be a viable method, although this measure is affected by several factors, including geography, cultivation practices, cultivar types, and harvest timing, among others [27].

### 3.3. Monosaccharide Composition Analysis of LBPs

Figure 2 illustrates the presence of eight monosaccharides in LBPs across various grades: Man, Rha, GlcA, GalA, Glc, Gal, Xyl and Ara, highlighting Ara and Gal as the primary components. This indicates a similarity in monosaccharide compositions among different fruit grades. Apart from the two main monosaccharide compounds, however, distinctions exist. Grade 1 fruit predominantly features Glc, while Grade 2 exhibits GlcA and GalA. Grade 3 features Glc and Rha, Grade 4 features GlcA and Glc, and Grade 5 presents a predominance of Glc and GalA.

Studies by Ma and Wu isolated water-soluble LBP-1 and LBP-3, also revealing Ara and Gal as predominant components with specific substance content ratios [28,29]. Comparatively, Wang investigated monosaccharide content and antioxidative stress protection in various *L. barbarum* varieties, identifying Gal, Ara and Glc as primary monosaccharide groups in LBPs [30]. The main monosaccharide composition observed in different grades of LBPs in this study aligns with broader research findings.

### 3.4. NMR Analysis

22 LBPs samples exhibiting feasible solubility in D_2_O were selected for ^1^H NMR analysis from the original 48 samples. Due to the weak correlation peaks in the ^1^H-^1^H COSY spectrum, two-dimensional experiments could not be successfully conducted for further structural analysis in this study. The proton spectra of grades 1 to 5 were shown in Figure 3A–E, indicating the characteristic polysaccharide signals within the *δ*5.50–3.00 ppm range. Generally, peaks within *δ*5.50–5.00 ppm are assigned to *α*-sugar residues, while those within *δ*5.00–4.50 ppm are assigned to *β*-sugar residues. Protons H-2 to H-6 in the glucoside rings typically assign peaks in the *δ*4.50–3.00 ppm range.

According to Xie’s review on polysaccharide NMR, the samples displayed sugar residues in both *α*- and *β*-configurations [31]. Signals at *δ*5.33 and *δ*5.29 indicated the presence of *α*-Rha and *α*-Glc/GlcA residues, respectively. Meanwhile, signals at *δ*5.27, *δ*5.15, and *δ*5.13 suggested the presence of *α*-GalA, *α*-Gal, and *α*-Man residues. *α*-Glc/Ara residues were also present at *δ*5.08, and *β*- Glc/Xyl residues at *δ*4.56. Notably, strong peaks at *δ*3.82–3.80 and *δ*3.34 ppm were assigned to Gal/Ara and Glc residues in the polysaccharides.

The ^1^H NMR data, combined with monosaccharide composition analysis from HPLC, revealed distinct monosaccharide signals. The signal in the range of *δ*3.82–3.80 ppm, while potentially containing contributions from other monosaccharides, is assigned primarily to arabinose (Ara) residues. This assignment is strongly supported by the HPLC data (Figure 2), which consistently identified Ara as one of the predominant monosaccharides in all LBPs samples. The consistent and high intensity of this peak across spectra is therefore confidently attributed to Ara. Although the relative intensities of the Ara, Gal, and Glu peaks were pronounced, it is important to note that the absence of certain monosaccharide signals in the hydrogen spectrum does not necessarily indicate their absence in the sample; their signals may be obscured by solvent peaks or other polysaccharide signals.

### 3.5. Correlation Analysis of Fruit Grades and ^1^H NMR Data

Figure 3F depicts the relative peak intensity superposition of samples across various grades within the *δ*5.50 to 3.00 ppm range, revealing an increasing trend in the Ara residue signal at *δ*3.82 to 3.80 ppm from grades 1 to 5. Concurrently, the area distribution of the Glc signal at *δ*3.34 did not display grade-specific characteristics.

Furthermore, the distribution and integrated area of the arabinose peaks in the NMR spectra across various samples were examined. A heat map displaying Ara peak integral values across different grades was generated (Figure 4). A discriminant function was established using Fisher’s method based on the Ara integral values of 22 samples [32]. Table 3 indicates that the first discriminant number exhibited an 86.8% variance interpretation capability with a high typical correlation coefficient of 0.996, signifying effective discrimination among sample groups.

Subsequently, a Wilks’ Lambda test (Table 4) applied to the discriminant function demonstrated its significance (*p* = 0.003, <0.01), validating its effectiveness in distinguishing fruit grade groups [33]. Utilizing this discriminant function (Figure 5), accurate classification was achieved (Table 5): six samples in the first class, four in the second, four in the third, four in the fourth, and four in the fifth class, resulting in a 100% accuracy rate.

Based on the analysis above, the distribution of chemical shifts and the integral values of the Ara peak in the ^1^H NMR spectra were identified as critical parameters for differentiating between various grades of fruits. In the spectra of different grades, signals corresponding to Man, Rha, Ara, Gal, Xyl, Glc, GlcA, and GalA were detected. Notably, the signals for Ara, Gal, and Glc were found to be more pronounced, aligning with the higher abundance of these monosaccharides.

Our results show that higher-grade fruits generally possess a higher polysaccharide content. More significantly, a detailed analysis of monosaccharide composition and ^1^H NMR data reveals a critical qualitative distinction: the relative abundance of arabinose (Ara) increases with higher fruit grades. This is not merely a compositional change but a fundamental indicator of superior bioactivity. The strong and grade-dependent Ara signal in the ^1^H NMR spectra, which allowed for accurate discriminant classification, underscores its role as a key quality marker.

### 3.6. In Vitro Hypoglycemic Activity Evaluation

The hypoglycemic activity of LBPs is highly dependent on their regulation of *α*-amylase and *α*-glucosidase enzymes, effectively managing the blood glucose levels. The IC_50_ values can quantify the effectiveness of a substance in inhibiting specific reactions, such as those of *α*-amylase and *α*-glucosidase, which are major contributing factors for controlling postprandial hyperglycemia [34]. Table 6 indicates that among various fruit grades, the IC_50_ values for *α*-amylase varied from 0.433 to 1.645 mg/mL, and for *α*-glucosidase, from 0.474 to 1.052 mg/mL. Higher grade fruits produced lower *α*-amylase IC_50_ values (5 < 4 < 3 < 2 < 1), yet higher *α*-glucosidase IC_50_ values (1 < 2 < 3 < 4 < 5). Although LBPs exhibited inhibitory effects on these enzymes, their activity was less potent compared to acarbose, a reference compound. Furthermore, a dose–response relationship was noted.

Partial least squares analysis was conducted to investigate the relationship between monosaccharide components and enzyme inhibition. The monosaccharide content data of LBPs samples were adopted as the independent variable X, and the inhibition rates of enzymes were used as the dependent variable Y (Figure 6). Monosaccharide components Glc, Ara, and GalA were key inhibitors of *α*-amylase (Figure 7), while Glc, Ara, Gal, and GalA played vital roles in inhibiting *α*-glucosidase (Figure 8), with the VIP values exceeding 1.0. These findings were supported by monosaccharide composition and ^1^H NMR data, suggesting Ara, Gal, and GalA as key components of LBPs.

The in vitro hypoglycemic activity further validates this structural superiority. The enhanced inhibitory effects against α-amylase and α-glucosidase observed in higher-grade fruits can be mechanistically explained by the unique structural roles of Ara, Gal, and GalA:

Synergistic Enzyme Inhibition: The notable presence of Ara and Gal indicates a high degree of branching in the arabinogalactan structures of LBPs [35]. As established in previous research, such highly branched, high-molecular-weight polysaccharides do not act as simple competitive substrates. Instead, they exert inhibition through steric hindrance, where their dense, entangled macromolecular structures physically block access to the active sites of digestive enzymes like α-amylase [36]. Furthermore, the multitude of exposed hydroxyl groups on these branched chains facilitates extensive hydrogen bonding with the enzymes [37], potentially inducing conformational changes that impair their catalytic function [38]. The contribution of GalA adds another layer of complexity, as its carboxylic acid groups can engage in electrostatic interactions, further enhancing the binding affinity and inhibitory potential [37].

Beyond Direct Inhibition—The Gut Microbiota Regulation: The in vitro enzyme assays capture only one facet of LBPs’ hypoglycemic mechanism. The high content of Ara and Gal in superior-grade fruits suggests these LBPs are ideal substrates for specific gut bacteria, such as *Bacteroides* and *Bifidobacterium*. Through fermentation, these microbes produce short-chain fatty acids (SCFAs) like acetate and propionate), of which contribute to improved systemic glucose regulation [39]. Therefore, the superior in vivo hypoglycemic effect of high-grade LBPs is likely a combined result of direct enzyme inhibition in the gut lumen and systemic metabolic regulation.

Understanding Enzyme Inhibition Patterns: The observed trend where higher-grade fruits showed stronger inhibition against α-amylase but slightly weaker inhibition against α-glucosidase is intriguing. This may reflect the differential sensitivity of these enzymes to the specific polymeric structure and monosaccharide arrangement of the polysaccharides. The active site of α-amylase, which cleaves internal α-1,4 glycosidic bonds in starch, might be more susceptible to blockade by the large, branched Ara/Gal-rich arabinogalactans [40]. On the other hand, α-glucosidase, which acts on terminal glucose residues, might be inhibited by a different set of structural features. This highlights that the activity of LBPs is not uniform, but depends on their detailed structure and how they interact with different enzymes.

In summary, our findings are consistent with the pattern that polysaccharides with blood sugar-lowering activity usually contain monosaccharides such as Ara, Gal, Glu, and Xyl, along with a small amount of GalA, Rha, and Fuc [41]. A new type of pectin polysaccharide PZMP3-2 was isolated from the wood jujube. Its main chain is a high galacturonic acid polysaccharide, and the side chains contain rhamnose, arabinose and galactose. This unique structure may confer its special biological activity [42].Our PLS-DA analysis validates that the hypoglycemic effect of LBPs is closely linked to the compositional abundance of its signature Ara, Gal, and GalA units. Further work will be conducted wo elucidate this structure–activity relationship.

## 4. Conclusions

In conclusion, this study moves beyond superficial phenotypic grading to establish a structure–activity-based quality assessment for *L. barbarum*. Our findings demonstrate that higher commercial grades are not only associated with smaller fruit size and higher polysaccharide content but, more importantly, with a superior polysaccharide quality characterized by a distinct monosaccharide signature.

The key findings are as follows:

Arabinose is a Signature Quality Marker: HPLC-DAD analysis revealed a consistent monosaccharide composition in LBPs, with Ara and Gal being the primary components. The increasing intensity of the Ara signal in ^1^H NMR spectra from lower to higher grades serves as an objective chemical marker for quality discrimination, successfully classifying fruits with impressive accuracy.

Signature Monosaccharides Determine Effectiveness: In vitro studies have revealed that crude LBPs inhibit both *α*-glucosidase and *α*-amylase, with Ara, Gal, and GalA being the main contributing monosaccharide components. They are the fundamental building blocks that promote hypoglycemic activity, potentially through the formation of highly branched arabinogalactan structures that inhibit digestive enzymes via steric hindrance and molecular interactions.

Dual Hypoglycemic Mechanism: The bioactivity of LBPs is a synergistic outcome of direct enzymatic inhibition in the small intestine and the prebiotic modulation of gut microbiota, contributing to systemic metabolic improvements.

Therefore, the quality of *L. barbarum* for functional food and medicinal applications should be evaluated based on its specific polysaccharide composition and structural features, with Ara, Gal, and GalA content serving as major standards. This research provides a firm theoretical foundation for refining commercial grading systems and for the targeted development of *Lycium barbarum*-based products for managing diabetes. Future work should focus on isolating these specific arabinogalactan fractions to confirm their structure–activity relationships and validate their in vivo efficacy and prebiotic potential.

## Figures and Tables

**Figure 1 foods-14-03862-f001:**
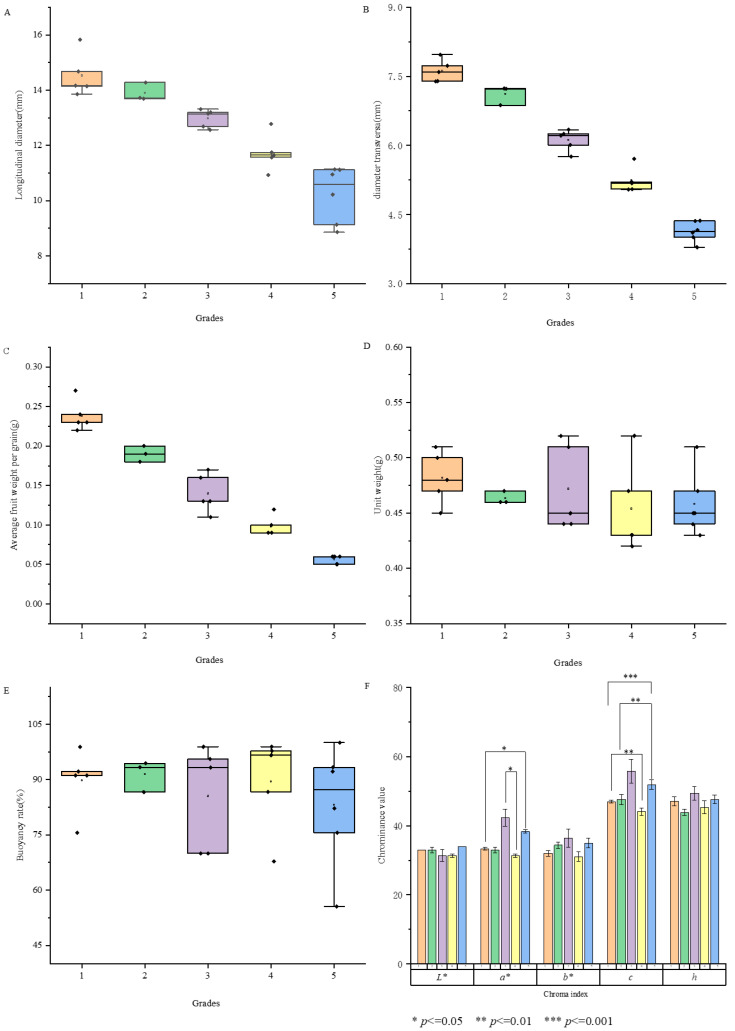
Longitudinal diameter (**A**), transverse diameter (**B**), average fruit weight (**C**), bulk density (**D**), floating rate (**E**) and color parameters (**F**) of different fruit grades.

**Figure 2 foods-14-03862-f002:**
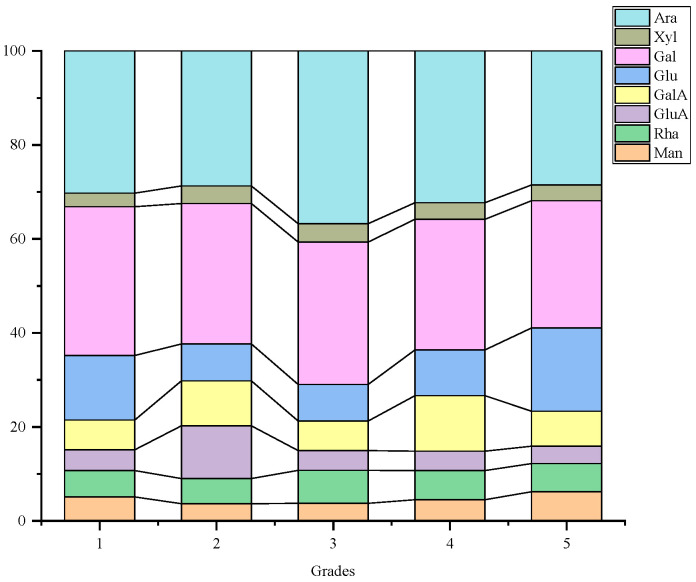
Accumulation map of monosaccharide composition of LBPs in different commodity grades.

**Figure 3 foods-14-03862-f003:**
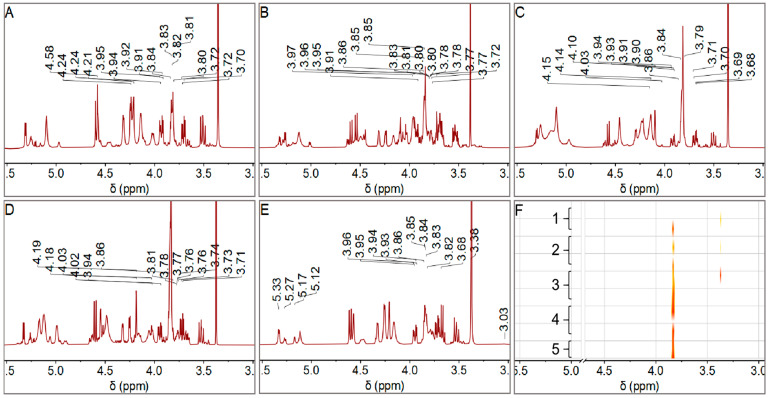
(**A**–**E**) ^1^H NMR (400.13 MHz, D_2_O, 300 K) spectra (*δ*3.00–5.50 ppm) of LBPs from Grades 1–5 fruits with solvent and impurity peaks omitted for clarity. (**F**) Stacked ^1^H NMR (400.13 MHz, D_2_O, 300 K) bitmap spectra (*δ*3.00–5.50 ppm) of LBPs samples across different grades. The color gradient from yellow to red indicates increasing peak intensity. Solvent peaks (*δ*4.79) are removed for clarity.

**Figure 4 foods-14-03862-f004:**
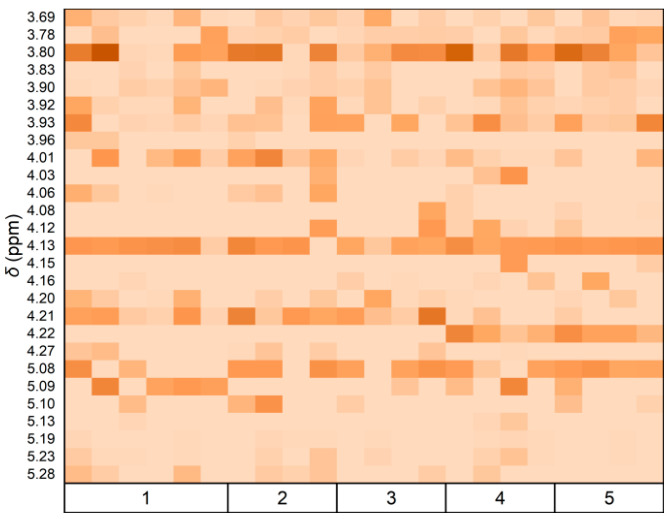
Heat map of arabinose ^1^H NMR (400.13 MHz, D_2_O, 300 K) peak integration magnitudes within *δ*3.00–5.50 ppm. The color gradient from light to dark indicates increasing peak integral values.

**Figure 5 foods-14-03862-f005:**
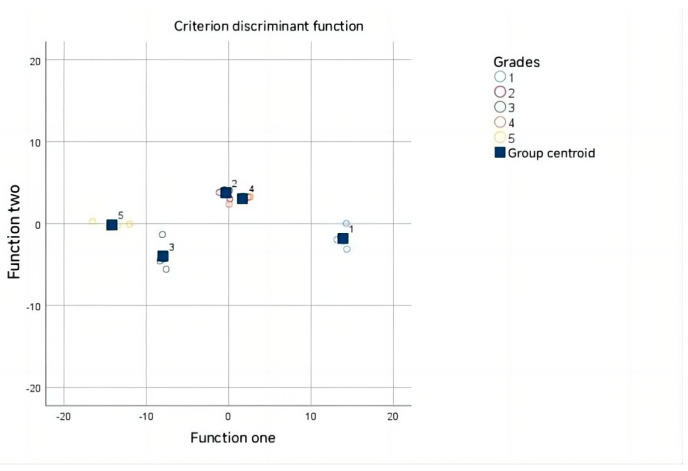
The discriminant function.

**Figure 6 foods-14-03862-f006:**
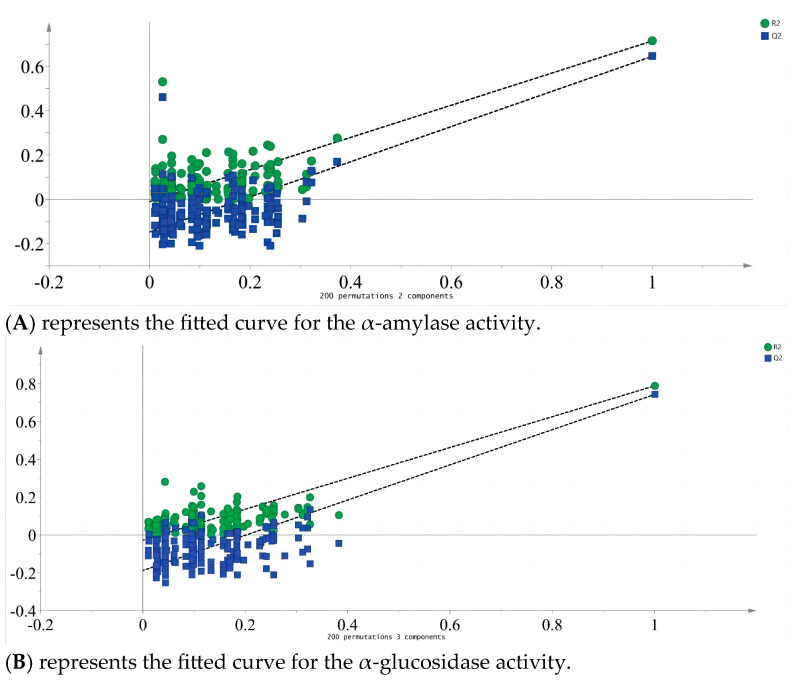
*α*-amylase and *α*-Glucosidase fit curves. (**A**) α-amylase; (**B**) α-glucosidase. (similarly hereinafter).

**Figure 7 foods-14-03862-f007:**
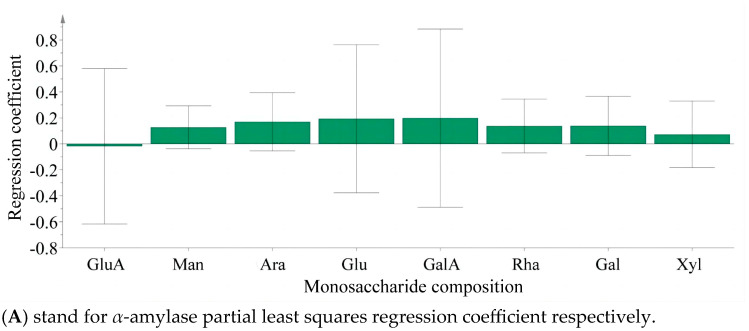
The content of monosaccharides in *Lycium barbarum* polysaccharide inhibited *α*-amylase and *α*-Glucosidase partial least squares regression coefficient.

**Figure 8 foods-14-03862-f008:**
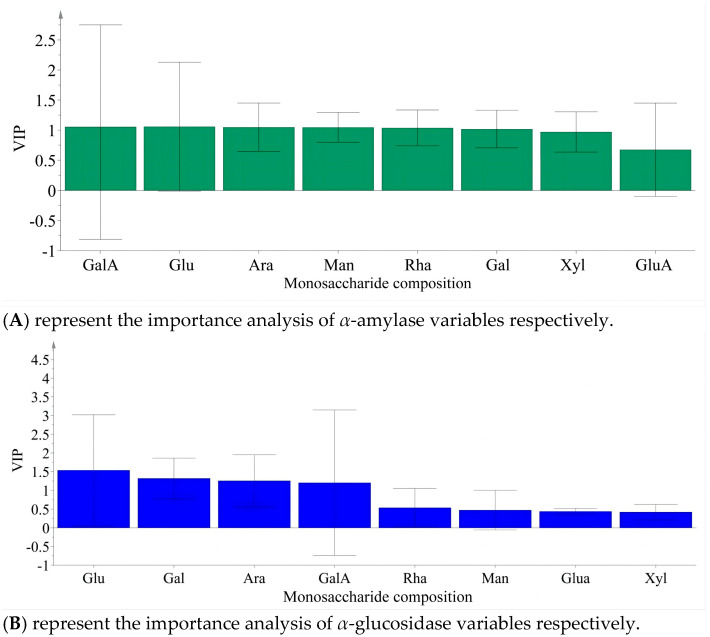
Analysis of the importance of monosaccharide content in LBPs polysaccharide in inhibiting *α*-amylase and *α*-Glucosidase variables.

**Table 1 foods-14-03862-t001:** Classification of *L. barbarum* Commercial Grades.

Grade	1	2	3	4	5
Fruit size (number of fruits per 50 g)	≤180	≤220	≤280	≤370	≤580

**Table 2 foods-14-03862-t002:** LBPs content across different grades.

Grade	1		2		3		4		5	
	Sample Number	Quality Score %	Sample Number	Quality Score %	Sample Number	Quality Score %	Sample Number	Quality Score %	Sample Number	Quality Score %
	1	3.20 ± 0.18 ^b^	3	3.55 ± 0.10 ^b^	6	3.41 ± 0.25 ^b^	7	4.55 ± 0.07 ^a^	9	3.41 ± 0.04 ^b^
	2	3.12 ± 0.14 ^b^	4	3.31 ± 0.27 ^b^	13	3.52 ± 0.02 ^b^	8	3.14 ± 0.25 ^cd^	33	3.61 ± 0.29 ^a^
	44	3.09 ± 0.15 ^b^	5	5.06 ± 0.04 ^a^	14	3.24 ± 0.10 ^c^	17	3.28 ± 0.19 ^c^	34	2.73 ± 0.14 ^d^
	10	2.37 ± 0.13 ^c^	11	2.93 ± 0.17 ^c^	15	5.69 ± 0.25 ^a^	19	4.51 ± 0.10 ^a^	40	2.54 ± 0.19 ^e^
	18	3.00 ± 0.34 ^b^	12	2.85 ± 0.10 ^c^	35	2.92 ± 0.31 ^c^	23	3.71 ± 0.22 ^b^	41	2.46 ± 0.16 ^f^
	21	3.62 ± 0.14 ^a^	16	3.35 ± 0.29 ^b^	36	2.65 ± 0.06 ^cd^	32	2.92 ± 0.55 ^d^	42	3.14 ± 0.07 ^c^
	29	2.85 ± 0.10 ^bc^	24	3.74 ± 0.20 ^b^	22	3.18 ± 0.08 ^c^	37	3.48 ± 0.38 ^bc^	43	2.57 ± 0.30 ^e^
	30	2.87 ± 0.18 ^bc^	25	2.21 ± 0.47 ^d^	27	1.94 ± 0.07 ^d^	38	2.65 ± 0.12 ^e^	48	3.11 ± 0.22 ^c^
	31	2.30 ± 0.12 ^c^	26	2.98 ± 0.73 ^c^	28	3.47 ± 0.14 ^b^	39	3.29 ± 0.02 ^c^	-	-
	20	3.04 ± 0.14 ^b^	45	3.46 ± 0.15 ^b^	46	3.64 ± 0.63 ^b^	47	3.34 ± 0.27 ^bc^	-	-
$\bar{x}$ ± s	-	2.95 ± 0.16	-	3.34 ± 0.25	-	3.37 ± 0.19	-	3.48 ± 0.21	-	2.94 ± 0.17

Note: “-” indicates not detected. Distinct superscript letters (a–f) within columns denote statistical significance (*p* < 0.05).

**Table 3 foods-14-03862-t003:** Function eigenvalues of typical discriminates based on integral values.

Function	Characteristic Value	Variance%	Accumulate%	Canonical Correlation
one	130.997 ^a^	86.8	86.8	0.996
two	10.361 ^a^	6.9	93.6	0.955
three	8.627 ^a^	5.7	99.4	0.947
four	0.967 ^a^	0.6	100.0	0.701

Notes: ^a^. The first four canonical discriminate functions were used in the analysis.

**Table 4 foods-14-03862-t004:** Lambda test based on Wilks.

Test of Function	Lambda for Wilks	Chi-Square	df	Sig.
one to four	0.000	102.541	68	0.004
two to four	0.005	53.714	48	0.265
three to four	0.053	29.412	30	0.496
four	0.508	6.766	14	0.943

**Table 5 foods-14-03862-t005:** Classification results.

Categories			Prediction Group Members					Total
			1.00	2.00	3.00	4.00	5.00	
Initial	Counting	1.00	6	0	0	0	0	6
		2.00	0	4	0	0	0	4
		3.00	0	0	4	0	0	4
		4.00	0	0	0	4	0	4
		5.00	0	0	0	0	4	4
	%	1.00	100.0	0.0	0.0	0.0	0.0	100.0
		2.00	0.0	100.0	0.0	0.0	0.0	100.0
		3.00	0.0	0.0	100.0	0.0	0.0	100.0
		4.00	0.0	0.0	0.0	100.0	0.0	100.0
		5.00	0.0	0.0	0.0	0.0	100.0	100.0

Note: 100.0% of the original grouped cases were correctly classified.

**Table 6 foods-14-03862-t006:** IC_50_ of *α*-amylase and *α*-glucosidase in crude LBPs from various grades.

Grades	1			2			3			4			5		
	Number	A	B	Number	A	B	Number	A	B	Number	A	B	Number	A	B
	1	0.61	0.55	3	0.64	0.62	6	0.59	0.52	7	0.66	0.64	9	0.45	0.63
	2	0.65	0.55	4	0.62	0.63	13	0.71	0.97	8	0.45	0.47	33	0.48	0.82
	10	0.50	0.60	5	0.62	0.73	14	0.77	0.49	17	0.51	0.67	34	0.46	0.83
	18	0.58	0.49	11	0.82	0.66	15	0.72	0.59	19	0.42	0.81	40	0.49	0.69
	20	0.52	0.63	12	0.60	0.66	22	0.50	0.68	23	0.46	0.96	41	0.53	0.80
	21	0.90	0.71	16	0.73	0.60	27	0.76	0.56	32	0.52	0.54	42	0.52	0.69
	29	1.65	0.72	24	0.79	0.64	28	0.59	0.66	37	0.78	0.81	43	0.48	0.84
	30	0.74	0.61	25	0.81	0.75	35	0.72	0.64	38	0.57	0.89	48	0.49	0.66
	31	0.97	1.05	26	0.81	0.73	36	0.89	0.72	39	0.46	0.58	-	-	-
	44	0.61	0.69	45	0.86	0.72	46	0.83	0.93	47	0.43	0.61	-	-	-
$\bar{x}$ ± s	-	0.77	0.66	-	0.73	0.67	-	0.71	0.68	-	0.52	0.70	-	0.49	0.74

Note: A and B refer to IC_50_ of *α*-amylase and *α*-glucosidase, respectively, where - indicates an infinite number of values.

## Data Availability

The original contributions presented in the study are included in the article/Appendix A, further inquiries can be directed to the corresponding author/s.

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
