# Peer review of "Assessment of Phenotypic Characteristics, Polysaccharide Composition, and Hypoglycemic Potential in Different Commercial Grades of *Lycium barbarum*: A Comprehensive Study Using HPLC and NMR"

_foods, 2025, doi:10.3390/foods14223862_

Round 1

Reviewer 1 Report

Comments and Suggestions for Authors

The study provides a thorough investigation into the phenotypic characteristics, polysaccharide composition, and hypoglycemic potential of various commercial grades of Lycium barbarum, employing a robust methodological framework integrating HPLC-DAD and 1H NMR spectroscopy. The phenotypic data are well-quantified, revealing significant differences in fruit size parameters across grades, although floating rate and bulk density showed no statistical divergence. The analysis of polysaccharide content and monosaccharide composition is comprehensive, identifying Ara, Gal, and GalA as dominant constituents and correlating these with biological activity. Particularly noteworthy is the use of ¹H NMR to establish discriminant functions based on arabinose peak intensities, enabling successful classification of grades with 100% accuracy. Furthermore, the inhibition assays against α-amylase and α-glucosidase are appropriately conducted and statistically evaluated, providing meaningful IC50 values. The application of multivariate analysis to associate specific monosaccharides with hypoglycemic activity strengthens the mechanistic insights. However, the study would benefit from a more detailed discussion of the biochemical pathways potentially modulated by Ara, Gal, and GalA, and their interactions with enzyme active sites. The inability to perform 2D NMR due to solubility limitations is a notable constraint and should be addressed through methodological refinement or sample pre-treatment. Additionally, the study lacks in vivo validation, which limits translational relevance. Overall, the manuscript offers valuable contributions to the standardization of L. barbarum quality assessment, though future work should aim to clarify the causal molecular mechanisms underlying the observed enzyme inhibition.

Questions:

  1. How do arabinose, galactose, and galacturonic acid interact with the active sites of α-amylase and α-glucosidase at the molecular level?
  2. Do Ara, Gal, and GalA act as competitive, non-competitive, or mixed-type inhibitors against these enzymes?
  3. Are there specific structural motifs within LBPs that enhance binding affinity to digestive enzymes?
  4. How does the degree of polymerization or branching in LBPs affect their inhibitory activity?
  5. What are the conformational dynamics of LBPs in aqueous solution, and how might these influence enzyme binding?
  6. Could Ara, Gal, and GalA indirectly modulate glucose metabolism pathways in vivo beyond enzymatic inhibition?
Comments on the Quality of English Language

The English is generally clear and scientific, but occasional grammatical inconsistencies and awkward phrasing reduce overall fluency.

Reviewer 2 Report

Comments and Suggestions for Authors

Although the idea of the work is good, the whole manuscript presents severe lacks. The objective of the work is described but it is weakly supported by the obtained results. The presentation of the materials and methods is not clear and a complete discussion is not present in the paper.

Here below, some of the main issues are listed:

Introduction

- You should give the meaning of LBPs. It is in abstract but you should explain it in the main text when it appears for the first time.

- What does supplementary table 1 data refer to? Your analysis or some reference? If it refers to your data you should put it on results sections. If it refers to a reference, you should only insert the citation.

- Line 67. You affirm that polysaccharides are responsible for goji berries biological activity. Polysaccharides are not the only responsible, since other molecules, i.e. polyphenols and others are present and responsible for biological activity. Change the sentence.

Materials and methods

- Line 94: change was with were

- Table 1: Is it referred to number of fruits per 50? Or other units?

- Line 126: petroleum ether has been used as extractant of polysaccharides or crude fat?

- Quantification of Polysaccharides Content is not properly explained. Give more information on the method.

- Line 160: What do you mean with process of heavy water dissolution and freeze-drying? All the method for NMR sample preparation is not clear

Results and discussion

- Grade 5 are characterized by lowest LBPs content, although they should have the highest one, basing on your sentence. Did authors explained this trend unexpected trend?

- Table 2: “- indicates an infinite number of values.”. What do authors mean with this expression? What do statistical data express in the table?

- Are data regarding monosaccharides composition expressed as absolute values or relative ones? Are they significatively different?

- Why do authors refer to peak intensity and nor peak areas for NMR analysis? Intensity can be affected by shim. Moreover, why did authors refer only to 3.8 ppm intensity? NMR assignment is nor clear. More details are needed. 3.8 ppm is commo also with other sugars. It is difficult to assign it only to Ara.

- What about fructose? It was not detected in any of the samples?

- It is not clear why only 22 NMR spectra are available. This is due to solubility in D2O? What about the other samples? No solubility was observed?

- Is there any correlation between HPLC and NMR data?

- You tested polysaccharides extract for biological activity but you talk about the action of single monosaccharides. How is it possible?

- Generally, discussion is not present in the section that is just limited to describe the obtained results. Few perspectives emerge from the text and few comparison with literature is present.

Conclusions

This section is a resume of the obtained results for each analysis. No sentences regarding the innovation of the study are present and no perspectives are mentioned.

Round 2

Reviewer 1 Report

Comments and Suggestions for Authors

The authors have made substantial and commendable improvements to the revised manuscript. The responses to the reviewer’s comments are thorough, scientifically grounded, and clearly integrated into the revised text. Notably, the discussion has been expanded to include plausible molecular mechanisms underlying the enzyme inhibitory effects of Ara, Gal, and GalA, supported by relevant literature. The limitations regarding 2D NMR and in vivo validation are now appropriately acknowledged. These revisions enhance the mechanistic depth, scientific rigor, and clarity of the study. The revised version demonstrates significant progress and addresses the critical concerns raised in the initial peer review.

Comments on the Quality of English Language

The English is generally clear and scientific

Reviewer 2 Report

Comments and Suggestions for Authors

The paper has been significatively improved and can be now considered for publication